# Essential Oil Blends: The Potential of Combined Use for Respiratory Tract Infections

**DOI:** 10.3390/antibiotics10121517

**Published:** 2021-12-10

**Authors:** Stephanie Leigh-de Rapper, Alvaro Viljoen, Sandy van Vuuren

**Affiliations:** 1Department of Pharmacy and Pharmacology, Faculty of Health Sciences, University of the Witwatersrand, 7 York Road, Parktown 2193, South Africa; Stephanie.derapper@wits.ac.za; 2Department of Pharmaceutical Sciences, Faculty of Sciences, Tshwane University of Technology, Private Bag X680, Pretoria 0001, South Africa; viljoenam@tut.ac.za; 3SAMRC Herbal Drugs Research Unit, Department of Pharmaceutical Sciences, Tshwane University of Technology, Private Bag X680, Pretoria 0001, South Africa

**Keywords:** antimicrobial, anti-inflammatory, inhalation, essential oils, combinations, synergy, toxicity, respiratory tract

## Abstract

This study investigated the potential efficacy of 369 commercial essential oil combinations for antimicrobial, anti-toxic and anti-inflammatory activity with the aim of identifying synergy among essential oils commonly used in combination by aromatherapists for respiratory purposes. Essential oil combinations were assessed for their antimicrobial activities using a panel of Gram-positive, Gram-negative, and yeast strains associated with respiratory tract infections. The antimicrobial activity was measured by determining the minimal inhibitory concentration (MIC) of microbial growth. The fractional inhibitory concentration index (ΣFIC) was calculated to determine the antimicrobial interactions between the essential oils in the combination. The toxicity of the essential oil combinations was tested *in vitro* using the brine shrimp lethality assay, the 3-(4,5-dimethylthiazol-2-yl)-2,5-diphenyltetrazolium bromide (MTT) assay on RAW 264.7 mouse macrophage cells and A549 lung cancer cell lines. In addition, an inflammatory response was evaluated measuring nitric oxide production. The essential oils, when in combination, demonstrated an increased antimicrobial effect, a reduction in toxicity and provided improved anti-inflammatory outcomes. Five distinct combinations [*Cupressus* *sempervirens* (cypress) in combination with *Melaleuca* *alternifolia* (tea tree), *Hyssopus* *officinalis* (hyssop) in combination with *Rosmarinus* *officinalis* (rosemary), *Origanum* *marjorana* (marjoram) in combination with *M.* *alternifolia*, *Myrtus* *communis* (myrtle) in combination with *M.* *alternifolia* and *Origanum* *vulgare* (origanum) in combination with *M.* *alternifolia*] were found to be the most promising, demonstrating antimicrobial activity, reduced cytotoxicity and improved anti-inflammatory effects. With the increased prevalence of respiratory tract infections and the growing antimicrobial resistance development associated with antimicrobial treatments, this study provides a promising complementary alternative for the appropriate use of a selection of essential oil combinations for use in the respiratory tract.

## 1. Introduction

Antimicrobial resistance, partly a consequence of inappropriate antibiotic use, has been continuously recorded globally and is considered one of the greatest challenges to global public health [1]. *Streptococcus pneumoniae*, a commonly isolated pathogen of the respiratory tract, has developed and spread resistance to antibiotics such as penicillins, with penicillin resistance often correlating to resistance to other additional antibiotics such as macrolides and tetracyclines [2,3]. A possible solution to growing antimicrobial resistance is the use of alternative and complimentary therapies which have been shown to elicit antimicrobial effects as well as holistically treat symptoms often caused by infections. The emergence of new antibacterial agents based on natural products is a priority in the field of scientific research with studies demonstrating the potential of these products against pathogens of multidrug resistant causes [4,5].

Essential oils are comprised of volatile, aromatic and complex chemical compounds such as alcohols, aldehydes, esters, ethers, ketones, phenols and terpenes. These essential oils are distilled from plant parts and commonly employed in aromatherapy. Within the scientific literature, essential oils have been extensively studied with relevance to respiratory conditions. A plethora of studies are available where essential oils have been studies for antimicrobial purposes as well as for inhalation in traditional practices and in maintaining basic health conditions [6,7,8,9,10,11]. Although there has been a surge in interest and encouraged direction in the field of essential oil research, much of the research published focuses on identifying the potential of a single essential oil [12]. Within aroma therapeutic use, however, essential oils are predominantly used in multiple combinations, as aromatherapy is based on the practice of combining multiple essential oils to achieve an enhanced therapeutic effect [13]. Insights into the use of complementary and alternative therapies, including aromatherapy, within developed countries has identified that $25 billion is spent annually by people actively seeking CAM treatments to prevent or treat ongoing infections [14]. This growth in the use of CAM treatments requires an equal growth in research for the evaluation of these products and practices for quality, safety and efficacy.

The aroma-therapeutic literature notes extensively the use of essential oils in combination for improved antimicrobial, anti-oxidative, anti-inflammatory, as well as antihistaminic effects [15]. Previous studies have demonstrated the therapeutic potential of some commercial and indigenous essential oils when tested in combination [16,17,18,19,20]. However, a lack of data still exists for the antimicrobial efficacy of a large number of commonly applied commercial essential oils used in combination. Inflammation is a physiological response by a host to regulate the body’s reaction to infections, injury or toxins [21]. Acute inflammation is initiated by tissue-resident innate immune cells that recognize pathogen-associated molecular patterns (PAMPs) or damage-associated molecular patterns (DAMPs) caused by infection or cell injury [22]. In the case of an infection caused by microbial invasion, initial recognition of infection is mediated by tissue-resident macrophages and mast cells. These cells induce inflammation through the production of a variety of inflammatory mediators, including chemokines, pro-inflammatory cytokines, vasoactive amines, eicosanoids and products of proteolytic cascades to enhance inflammatory signals and recruit more immune cells [23]. The release of cytokines, such as interferon-γ (INF-γ), TNF-α, and interleukin-1 (IL-1), as well as exposure to microbial products such as LPS further stimulate macrophages to produce nitric oxide (NO) [24]. The immediate effect of these mediators is to cause local vasodilation as a means of providing plasma proteins and leukocytes (mostly neutrophils), normally held in circulation, to the extravascular tissues at the site of infection. Once at the site of infection, neutrophils become activated either by direct contact with pathogens or through the actions of cytokines secreted by tissue-resident cells [23]. The neutrophils then attempt to kill the invading pathogen by releasing the toxic contents of their granules [25]. The production of NO by macrophages further contributes to the elimination of invading pathogens by inducing the cytotoxic action of these macrophages. When the inflammatory response has resulted in the elimination of the infectious agents, a resolution and repair phase commences to reduce inflammation [26]. If the inflammatory response does not result in the elimination of the causative pathogen, the inflammatory process persists, leading to a chronic state of inflammation. The characteristics of this inflammatory state differ depending on the effector class of the T cells that are present [23]. The further influx of immune cells and increased inflammatory response can cause a reduction in gas exchange due to fluid accumulation, and further result in damage to the lungs, resulting in severe respiratory infection [27].

Essential oils have been well studied, and demonstrate anti-inflammatory effects [28,29,30,31]. Essential oils with anti-inflammatory activity have been shown to reduce swelling and edema associated with respiratory infections, thus reducing symptoms such as wheezing, congestion and difficulty in breathing. Despite this, essential oils are commonly used in combination in aromatherapy, and very little literature exists to support this application of essential oils for anti-inflammatory effects. Given the importance of the inflammatory process in infections of the respiratory tract, it is important to determine the potential for essential oils used in combination to influence this infection process.

Essential oils have also been associated with high levels of toxicity, especially when compared to other natural products [15]. Essential oils are registered on the United States Food and Drug Authority (US, FDA) list of ‘generally recognized as safe’ (GRAS) products as individual agents but not as combinations [15]. The activity between blended essential oils has the potential to increase toxicity or suppress these effects. Small doses of essential oils are capable of eliciting toxic effects, especially when inhaled [32]. Despite these findings, essential oils remain the most popular form of complementary medicine and continue to be used in combination via dermal or inhalation application. It is a result of this lack of data to substantiate the use of essential oils in combination for antimicrobial and anti-inflammatory effects, as well as the increased risk of toxicity associated with essential use when combined, that further research is needed to confirm the therapeutic potential for use of essential oils via the respiratory tract. This study, therefore, aims to determine the antimicrobial, anti-inflammatory and toxicity levels of commercial essential oil combinations commonly applied in aromatherapy for respiratory tract infections.

## 2. Results and Discussion

### 2.1. Antimicrobial Analysis of Essential Oils

The MIC values of the 49 individual commercial essential oils were investigated in our previous study [33]. Of the 369 combinations investigated (Appendix A), 41.6% displayed noteworthy activity against the nine respiratory pathogens investigated. Of the combinations determined to be noteworthy in antimicrobial effect, 27.3% elicited strongly noteworthy antimicrobial effects of less than 0.50 mg/mL, while 72.3% of combinations displayed moderately noteworthy effects between 0.50 and 1.00 mg/mL. Figure 1 summarizes the percentage of essential oils that were found to have a noteworthy antimicrobial activity alone (a) and in combination (b).

From these results, it is clear that the essential oils show the potential to demonstrate improved antimicrobial efficacy when in combination. As the focus was on finding the combinations with the highest antimicrobial efficacy, the best results observed for the 369 investigated combinations (1:1) against respiratory tract pathogens are summarized in Table 1.

The pathogen most sensitive to the effects of the essential oil combinations was *Cryptococcus neoformans* (average MIC of 0. 49 mg/mL). The antimicrobial effect of essential oils against fungal pathogens has been well studied against the test micro-organisms investigated here-in [34,35]. The genus of *Cryptococcus* is an important cause of opportunistic fungal infection in severely immunocompromised patients with the primary sites of infection including the lungs [36]. The lowest MIC value determined for the combination of essential oils against *C. neoformans* was 0.06 mg/mL for 24 combinations including the combination of *Citrus limon* (lemon) and *Santalum austrocaledonicum* (sandalwood), which were shown to have broad-spectrum activity. The essential oils of *C. limon* and *S. austrocaledonicum* have demonstrated antimicrobial activity with marked antifungal effects [33,37,38,39].

The respiratory tract pathogens most commonly neglected in research include *Haemophilus influenzae* and the *Mycobacterium* and *Streptococcal* species [15]. Of these neglected pathogens, *M. smegmatis* demonstrated the greatest sensitivity to essential oils with 249 combinations proving synergistic and no antagonism noted (Appendix A). The use of essential oils against tuberculosis has been commonly utilized with practices ranging from the inhalation of essential oils [40], to the application of essential oils within a diffuser to reduce bacterial loads at patient’s bedsides [41,42]. A commonly used essential oil combination for the reduction of *Mycobacterium* air content include a mixture of *Melaleuca quinquenervia* (niaouli), *Eucalyptus smithii* (eucalyptus), *Myrtus communis* (myrtle), *Abies balsamea* (fir balsam), *Melaleuca alternifolia* (tea tree), *Pelargonium asperum* (geranium) and *Mentha piperita* (peppermint) essential oils [42]. Of these essential oils studied in 1:1 ratios, the ΣFIC values ranged from 0.42 to 0.50, further supporting their synergistic potential for use in combination. Essential oils have demonstrated moderate antimicrobial effects against *H*. *influenzae* and the *Streptococcal* pathogens commonly associated with respiratory infections [43,44,45,46,47,48,49]. These findings were supported in the current study.

Several additive and synergistic combinations could be observed with 57.1% of combinations demonstrating these effects. Antagonism was noted in only 5.4% of combinations. The combination of *Coriandrum sativum* (coriander) and *Cinnamonum zeylanicum* (cinnamon) showed the broadest spectrum of antimicrobial effect with noteworthy effect against eight of the nine pathogens studied with synergistic or additive effects against five of the nine pathogens. These two essential oils have been used since antiquity for their antimicrobial effects [50,51]. An additional two combinations, namely *C. zeylanicum* in combination with *Zingiber officinale* (ginger) and *Citrus bergamia* (bergamot) in combination with *Rosmarinus officinalis* (rosemary) have shown the greatest number of synergistic interactions with synergy identified against six of the nine pathogens investigated.

### 2.2. Brine-Shrimp Lethality Assay (BSLA)

All essential oils and their 1:1 combinations were screened at 1 mg/mL, where toxicity was considered where the essential oils induced mortality at a percentage greater than 50% [52]. Of the combinations investigated (Appendix A), the toxicity results of the single oils (a) and combinations (b) demonstrating the least toxicity are summarized in Figure 2.

Approximately 31.0% of the combinations studied demonstrated non-toxic effects against the brine shrimp. From these results, it appears that the essential oils show the potential to quench toxicity of individual essential oils when used in combination. Selected combinations (Table 2) which were based on the good antimicrobial activity indicated that although the combinations remain non-toxic, 76.0% of essential oils increase in toxicity when combined. The essential oil toxicity after 48 h when studied individually increased from 20.4% to 42.5% once combined. The combination of *Syzygium caryophyllata* (clove) with *C. zeylanicum*, or with *Thymus vulgaris* (thyme) showed the greatest synergistic (non-toxic effects) against brine shrimp with an ΣFIC of 0.03; followed by the combination of *S. caryophyllata* with *Cymbopogon citratus* (lemongrass) with an ΣFIC of 0.05.

While the BSLA serves as an excellent tool to identify toxicity, a more sensitive distinction of activity is the testing against cell lines [53]. Hence, the BSLA was used to identify combinations of interest, which were then followed by the MTT assay using RAW 264.7 macrophages and A549 lung cell lines to substantiate the results of the BSLA as the latter study considers the metabolic activity of the cells.

### 2.3. Inhibition of LPS-Induced Macrophage Activation

From the results derived in the combined antimicrobial assays, and the BSLA, 24 essential oil combinations were found to be non-toxic and have broad-spectrum antimicrobial activity, and were therefore selected for further anti-inflammatory studies. The anti-inflammatory effect was studied by measuring the production of a single inflammatory marker, nitric oxide (NO). As inflammation is a very complex process, this assay was applied to determine the preliminary potential of essential oils for anti-inflammatory effects. Additional toxicity studies against lung tissue were undertaken to further identify the potentially toxic effects of essential oils commonly inhaled for respiratory conditions.

Statistical analysis was undertaken using StatSoft Inc. (Tulsa, OK, USA) (2004) STATISTICA (Data Analysis Software System), Version 7 software. An area under the curve (AUC) and Receiver Operator Characteristic (ROC) study was performed to determine the optimum cut-off point for the production of NO. This was created taking into consideration cell cytotoxicity. The cut-off for essential oils demonstrating anti-inflammatory activity, relevant to cell cytotoxicity, was therefore determined as 2.97 μM nitrite production. The majority of essential oil combinations studied (62.5%), demonstrated an anti-inflammatory effect when compared to LPS, with anti-inflammatory activity demonstrated as 1.98 to 2.97 μM nitrite production compared to the control, LPS at 4.90 μM. Figure 3 summarizes the percentage of essential oils with anti-inflammatory effect when tested independently (a) and when in combination (b).

From these results, essential oils showed an increased potential for anti-inflammatory effects when used in combination. Table 3 represents the results for the essential oils investigated individually and the 24 selected combinations (1:1) for anti-inflammatory effect.

The essential oil combination with the greatest anti-inflammatory effect was *Cupressus sempervirens* (cypress) with *Hyssopus officinalis* (hyssop) having a nitrite production as low as 1.98 μM. Previous studies have shown the anti-inflammatory potential of these essential oils [54], however, little is known of the combined anti-inflammatory effects of these essential oils identified in this study [15]. To confirm the absence of cytotoxicity caused by exposure of the RAW macrophage cells to these essential oil combinations, leading to a reduced NO production, an MTT assay was run concurrently. The results of the MTT assay are provided in Table 4.

The results of the MTT assay indicate that cytotoxicity is evident in 50% of the essential oil combinations towards the RAW 264.7 cells, providing reduced confidence in the anti-inflammatory results associated to these combinations as these may be meaningfully confounded by cell death. Of the combinations studied, three combinations demonstrated cell viability above 80% and NO production less than 4.00 μM, namely *Origanum marjorana* (marjoram) with *M. alternifolia* (3.05 uM, 90.14%), *Origanum vulgare* (origanum) with *M. alternifolia* (3.87 uM, 86.31%), and *C. sempervirens* with *M. alternifolia* (3.49, 83.83%).

Numerous studies support the anti-inflammatory activity of *M. alternifolia* essential oil both *in vitro* and *in vivo* [55,56,57,58]. Studies have shown the potent effects of *M. alternifolia* essential oil in inhibiting the production of the inflammatory mediators tumour necrosis factor alpha (TNF-α), interleukin-1β (IL-1β), IL-10, and that of prostaglandin E2 [58]. Previous studies support the anti-inflammatory effect of *Origanum* essential oils [59,60,61]. Research on the anti-inflammatory effect of the essential oils *O. marjoram* and *O. vulgare* determined marked anti-inflammatory effect of these oils against TNFα, IL-1β, IL-6, IL-10 and inhibition of cyclooxygenase-2 [59,61]. These findings thus support those provided in literature of the essential oils *M. alternifolia*, *O. marjoram* and *O. vulgare* to produce anti-inflammatory effects and further demonstrate increased efficacy in combination.

### 2.4. Cytotoxic Effects of Essential Oils against A549 Lung Cancer Cells

To evaluate the cytotoxic effects of the essential oils and essential oil combinations against cells of the lung, the A549 cell line was exposed over a 48 h period and examined by the MTT assay. The results for 24 essential oil combinations are provided in Table 5.

The essential oils retained cell viability between 84.2% and 100.0%, while the combinations showed A549 cell viability between 80.4% and 99.6%. Consequently, all essential oils and their combinations showed non-toxic effects against the lung A549 cell lines. These findings further support those determined in the BSLA and provide an initial indication for the safe therapeutic potential of essential oils when used in the respiratory tract. In the majority (75.0%) of the combinations studies, additive effects were demonstrated when essential oils were tested against A549 cell lines. Among the tested essential oil combinations, *Lavandula angustifolia* (lavender) and *Citrus aurantifolia* (lime) exhibited the lowest (i.e., best effect in combination) additive effects (ΣFIC = 0.81) to the A549 cells compared to the other combinations.

## 3. Materials and Methods

The methods undertaken in this study aim to determine the best essential oil combination for use in the respiratory tract. All studies were undertaken in triplicate on consecutive days. Varying methods were applied to identify the most suitable combination of essential oil from the initial 369 combinations studied. An overview of the study design is provided in Figure 4.

### 3.1. Essential Oil Combination Selection, Procurement and Chemical Characterization

A selection of 49 essential oils were obtained from international fragrance companies Robertet© (Grasse, France), Burgess and Finch (Cape Town, South Africa), PranaMonde (Belgium) and Escentia Oils (Gauteng, South Africa). The essential oil combinations were selected based on the frequency of citation in the aroma-therapeutic literature available to the layman, with specified recommendation in treating respiratory tract infections [62,63,64,65,66,67,68]. The chemical analysis for all oils in this study has previously been characterized using gas chromatography coupled with a mass spectrometer (GC/MS) and is reported in our earlier studies [33].

### 3.2. Antimicrobial Analysis

Microbial cultures were selected based on their relevance to respiratory infections and included the Gram-positive strains *Staphylococcus aureus* (ATCC 25924), *Streptococcus agalactiae* (ATCC 55618), *Streptococcus pneumoniae* (ATCC 49619) and *Streptococcus pyogenes* (ATCC 12344) and the Gram-negative strains *Haemophilus influenzae* (ATCC 19418), *Klebsiella pneumoniae* (ATCC 13883) and *Moraxella catarrhalis* (ATCC 23246). The non-pathogenic *Mycobacterium* strain *M*. *smegmatis* (ATCC 19420) and yeast strain *Cryptococcus neoformans* (ATCC 14116) were also selected for analysis. All cultures were prepared as per Leigh-de Rapper et al. [33]. A waiver for the use of these micro-organisms was granted by the University of the Witwatersrand Human Research Ethics Committee (Reference W-CJ-160720-2).

The broth microdilution method described by de Rapper et al. [18] was used to quantify the antimicrobial inhibitory activity of the selected essential oils. The antimicrobial activities of the essential oils independently have been previously investigated [33]. The combinations of essential oils were undertaken at 1:1 ratios. The micro-titre plates were prepared aseptically [33]; however, a stock concentration of each essential oil (32 mg/mL in acetone) in the 1:1 combination was added to the first row at a volume of 50 µL per oil. The MIC values were recorded and the fractional inhibitory concentration index (ΣFIC) was calculated according to the review by van Vuuren and Viljoen [13] using the following equations;
$$\mathrm{FIC}\;\left(i\right)=\frac{\mathrm{MIC}\;\mathrm{of}\;\left(a^{*}\right)\;\mathrm{combined}\;\mathrm{with}\;\left(b^{*}\right)}{\mathrm{MIC}\;\mathrm{of}\;\left(a\right)\;\mathrm{independently}}$$
$$\mathrm{FIC}\;\left(\mathrm{ii}\right)=\frac{\mathrm{MIC}\;\mathrm{of}\;\left(b\right)\;\mathrm{combined}\;\mathrm{with}\;\left(a\right)}{\mathrm{MIC}\;\mathrm{of}\;\left(b\right)\;\mathrm{independently}}$$
* where (a) is the MIC of the one oil in the combination and (b) is the MIC of the other.

From these values the FIC index was calculated as: ΣFIC = FIC (i) + FIC (ii). The ΣFIC for each oil combination was interpreted where an ΣFIC value of ≤0.5 is indicative of synergy, an ΣFIC value of >0.5–1.0 is additive, an ΣFIC of >1.0–≤ 4.0 indicates indifference and an ΣFIC value of >4.0 indicates antagonism [13].

### 3.3. Brine-Shrimp Lethality Assay (BSLA)

The BSLA was used to quantify the toxic effects of the selected essential oils [69]. A volume of 400 μL salt water containing on average 40–60 live brine-shrimp was added to each well of a 48 well micro-titre plate. Thereafter, 400 μL of essential oil sample (essential oil or a combination of essential oils (1:1), all diluted in 1% dimethyl sulphoxide (DMSO)) was added to wells. All samples were tested for toxicity at a concentration of 1 mg/mL, since a concentration above 1 mg/mL not resulting in brine-shrimp death was considered non-toxic for the assay [52]. The negative control consisted of 32 g/L salt water, a solvent control of 1 mg/mL DMSO in water and the positive control consisted of 1.60 mg/mL potassium dichromate (Fluka). The micro-titre plates were observed under a light microscope (Olympus) (40 × magnification) immediately after sample addition (at time 0) for any dead brine-shrimp, which would be excluded from the percentage mortality calculations [70]. Dead brine-shrimp were then counted after 24 and 48 h. Thereafter, a lethal dose of 50 μL of glacial acetic acid (100% *v*/*v*; Saarchem, Maharashtra, India) was added to each well and a total dead brine-shrimp count undertaken. Samples providing a percentage mortality greater than 50% were considered toxic [52].

### 3.4. Anti-Inflammatory and MTT Cytotoxicity Assay

Forty-nine essential oils and twenty-four essential oil (1:1) combinations were selected for study based on the favorable results from the MIC assay and BSLA. The anti-inflammatory activity of these essential oil combinations were assessed by measuring the inhibition of nitric oxide (NO), an inflammatory mediator, production in LPS-activated RAW 264.7 macrophages. The control, lipopolysaccharides (LPS), is a potent inducer of nitric oxide synthase (iNOS) and concomitant NO producer, hence stimulating the signaling cascade that contributes to the inflammatory response in RAW 264.7 macrophage cells. Simultaneous evaluation of cell viability, using an MTT assay, was used to confirm the absence of cytotoxicity of these combinations. The essential oils were prepared in DMSO to a stock concentration of 10% and further diluted in culture medium to a concentration of 0.02%. RAW 264.7 mouse macrophage cells (Cellonex, South Africa) were cultured in RPMI complete medium, comprised of RPMI1640 supplemented with 10% fetal calf serum (GE Healthcare Life Sciences, (Logan, UT, USA)) in a humidified 5% CO_2_ incubator at 37 °C. Cells were seeded into 96-well plates at a density of 1 × 10^5^ cells per well and allowed to attach overnight. Spent culture medium was replaced with 50 µL of the samples (diluted in RPMI complete medium to 0.02%) or complete medium only (control and LPS control; Sigma-Aldrich, (St. Louise, MO, USA)) or aminoguanidine (AG; positive control; Sigma-Aldrich, (St. Louise, MO, USA)). To assess the anti-inflammatory activity, 50 μL of 1 µg/mL LPS in RPMI complete medium was added to all wells except controls, which received 50 μL medium. Final concentrations were 0.01% for essential oils, 500 ng/mL for LPS and 100 μM for AG. Cells were incubated for a further 24 h. To quantify nitric oxide (NO) production, 50 μL of the spent culture medium was transferred to a new 96-well plate and 50 μL Griess reagent (Sigma-Aldrich, (St. Louise, MO, USA) was added. Samples were incubated at room temperature for 10 min before absorbance was measured at 540 nm and the results expressed relative to the appropriate untreated LPS control. A standard curve using sodium nitrite dissolved in culture medium was used to determine the concentration of NO in each sample. To confirm the absence of toxicity as a contributory factor, cell viability was assessed using 3-(4,5-dimethylthiazol-2-yl)-2,5-diphenyltetrazolium bromide (MTT). This was done by replacing the remaining medium in each well with medium containing 0.5 mg/mL MTT and further incubation for 30 min at 37 °C. Thereafter, MTT was removed and 100 μL DMSO was added to each well to solubilize the formazan crystals. Absorbance was read at 540 nm using a BioTek^®^ PowerWave XS spectrophotometer (Winooski, VT, USA).

### 3.5. Data Analysis

Data was analysed using StatSoft Inc. (Tulsa, OK, USA) (2004) STATISTICA (Data Analysis Software System), Version 7. Standard deviations were calculated to determine standard error of the mean. Further, logistic regression using an area under the curve (AUC) and Receiver Operator Characteristic (ROC) were applied to determine the optimum cut-off point for the production of NO by macrophages relevant to cell cytotoxicity.

## 4. Conclusions

This study elucidates the promising strategy of combining essential oils for use in the respiratory tract. This study is the first to report on the majority of these essential oil combinations against the pathogens of the respiratory tract as well as anti-inflammatory and cytotoxic effects on lung cell lines. Based on the findings of this study, five distinct combinations of essential oils have been determined as the most promising for use in the respiratory tract based on the antimicrobial, cytotoxic and anti-inflammatory effects. These combinations include; *C. sempervirens* in combination with *M. alternifolia*, *H. officinalis* in combination with *R. officinalis*, *O. marjorana* in combination with *M. alternifolia*, *M. communis* in combination with *M. alternifolia,* and *O. vulgare* in combination with *M. alternifolia*. Future studies are underway which examine the optimum ratios at which these essential oil combinations should be combined to provide a holistic blend that would be non-toxic and elicit the most favorable antimicrobial and anti-inflammatory activity.

## Figures and Tables

**Figure 1 antibiotics-10-01517-f001:**
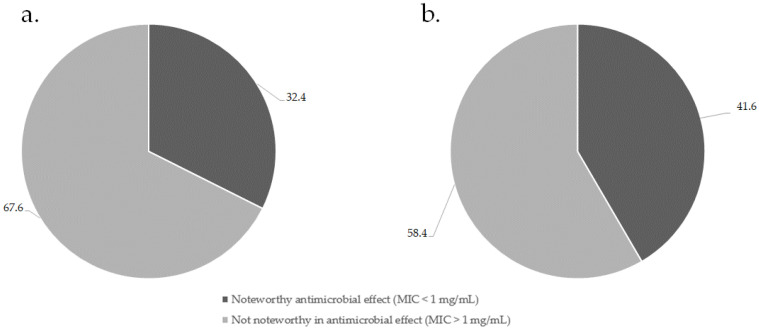
Summary of noteworthy essential oil antimicrobial activity alone (**a**) and when in combination (**b**).

**Figure 2 antibiotics-10-01517-f002:**
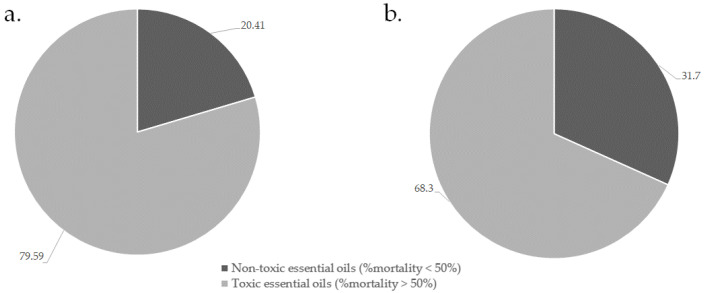
Summary of essential oil toxic effects alone (**a**) and when in combination (**b**) against brine shrimp.

**Figure 3 antibiotics-10-01517-f003:**
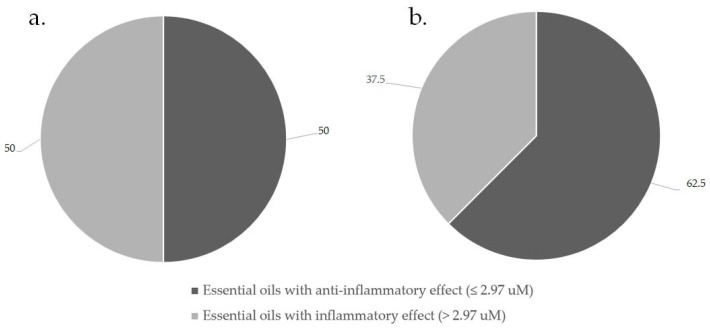
Summary of essential oil anti-inflammatory effects alone (**a**) and when in combination (**b**).

**Figure 4 antibiotics-10-01517-f004:**
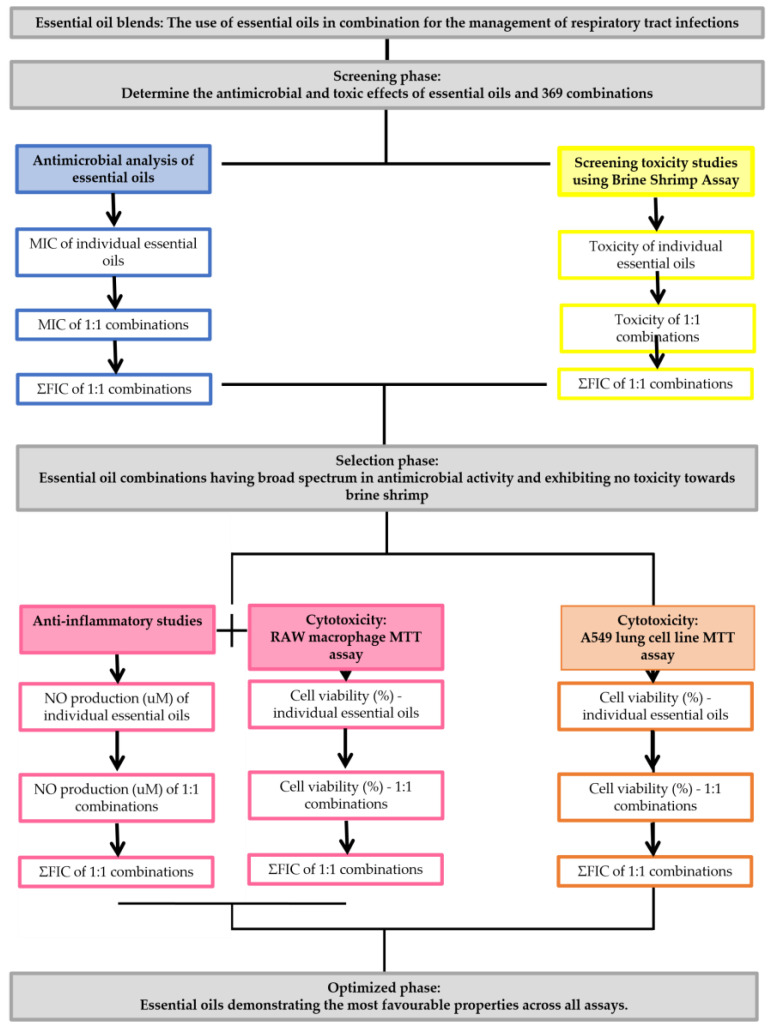
Summary of study workflow.

**Table 1 antibiotics-10-01517-t001:** The mean MIC (*n* = 3) with standard deviation in brackets and ΣFIC values of the essential oil combinations investigated against pathogens of the respiratory tract.

Essential Oil Combinations (Common Name in Brackets)		Mean MIC Value (mg/mL) (*n* = 3) and ƩFIC																	
		*Staphylococcus aureus*		*Streptococcus agalactiae*		*Streptococcus pneumoniae*		*Streptococcus pyogenes*		*Mycobacterium smegmatis*		*Haemophilus influenzae*		*Klebsiella pneumoniae*		*Moraxella catarrhalis*		*Cryptococcus neoformans*	
		(ATCC 25924)		(ATCC 55618)		(ATCC 49619)		(ATCC 12344)		(ATCC 19420)		(ATCC 19418)		(ATCC 13883)		(ATCC 23246)		(ATCC 14116)	
Essential Oil 1	Essential Oil 2	MIC *	ƩFIC **	MIC *	ƩFIC **	MIC *	ƩFIC **	MIC *	ƩFIC **	MIC *	ƩFIC **	MIC *	ƩFIC **	MIC *	ƩFIC **	MIC *	ƩFIC **	MIC *	ƩFIC **
*Amyris balsamifera* (amyris)	*Styrax benzoin* (benzoin)	**0.25 (±0.00)**	** *0.28* **	**0.25 (±0.00)**	** *0.31* **	**0.75 (±0.35)**	3.09	**0.19 (±0.00)**	** *0.31* **	2.00 (±0.00)	1.25	**0.13 (±0.00)**	*0.53*	1.50 (±0.71)	*0.75*	**0.50 (±0.00)**	** *0.38* **	**0.50 (±0.00)**	1.17
*Cinnamonum zeylanicum* (cinnamon)	*Elettaria cardamonum* (cardamom)	**0.50 (±0.00)**	** *0.50* **	**1.00 (±0.00)**	1.25	2.00 (±0.00)	*1.00*	**0.75 (±0.35)**	1.69	**1.00 (±0.00)**	** *0.31* **	2.00 (±0.00)	1.67	**0.50 (±0.00)**	** *0.50* **	2.00 (±0.00)	2.50	**0.13 (±0.00)**	** *0.13* **
	*Zingiber officinale* (ginger)	**0.50 (±0.00)**	** *0.50* **	**0.50 (±0.00)**	*0.75*	**1.00 (±0.00)**	** *0.50* **	**0.19 (±0.00)**	** *0.47* **	4.00 (±0.00)	2.00	**0.50 (±0.00)**	** *0.42* **	**0.50 (±0.00)**	** *0.31* **	2.00 (±0.00)	3.00	**0.13 (±0.00)**	** *0.21* **
*Citrus bergamia* (bergamot)	*Cupressus sempervirens* (cypress)	4.00 (±0.00)	*0.92*	2.00 (±0.00)	*1.00*	**1.00 (±0.00)**	** *0.31* **	6.00 (±2.83)	2.00	**1.00 (±0.00)**	** *0.25* **	1.50 (±0.71)	* **0.50** *	2.00 (±0.00)	*0.83*	2.00 (±0.00)	** *0.25* **	**1.00 (±0.00)**	1.50
	*Rosmarinus officinalis* (rosemary)	2.00 (±0.00)	** *0.38* **	**1.00 (±0.00)**	** *0.50* **	2.00 (±0.00)	*1.00*	8.00 (±0.00)	5.33	2.00 (±0.00)	** *0.50* **	**0.75 (±0.35)**	** *0.17* **	**0.75 (±0.35)**	** *0.31* **	**1.00 (±0.00)**	** *0.31* **	1.50 (±0.71)	3.75
*Citrus limon* (lemon)	*Santalum austrocaledonicum* (sandalwood)	**0.38 (±0.18)**	** *0.27* **	**0.13 (±0.00)**	** *0.28* **	**0.25 (±0.00)**	** *0.13* **	**0.19 (±0.00)**	1.55	2.00 (±0.00)	1.58	**0.50 (±0.00)**	2.03	4.00 (±0.00)	*0.92*	**1.00 (±0.00)**	2.50	**0.06 (±0.00)**	** *0.19* **
*Coriandrum sativum* (coriander)	*Cinnamonum zeylanicum* (cinnamon)	**0.50 (±0.00)**	*0.75*	**0.13 (±0.00)**	* **0.19** *	**1.00 (±0.00)**	*0.75*	**0.38 (±0.18)**	1.50	**0.50 (±0.00)**	* **0.25** *	**0.50 (±0.00)**	*1.00*	**1.00 (±0.00)**	1.50	2.00 (±0.00)	2.67	**0.13 (±0.00)**	2.08
*Cupressus sempervirens* (cypress)	*Hyssopus officinalis* (hyssop)	2.00 (±0.00)	*0.83*	2.00 (±0.00)	*1.00*	**1.00 (±0.00)**	* **0.31** *	1.50 (±0.71)	*1.00*	1.50 (±0.71)	*0.38*	2.00 (±0.00)	* **0.46** *	**1.00 (±0.00)**	*0.50*	2.00 (±0.00)	*0.63*	**0.50 (±0.00)**	2.50
	*Lavandula angustifolia* (lavender)	2.00 (±0.00)	1.33	2.00 (±0.00)	*1.00*	1.50 (±0.71)	*0.84*	8.00 (±0.00)	2.33	2.00 (±0.00)	*0.75*	4.00 (±0.00)	1.33	**1.00 (±0.00)**	*0.58*	2.00 (±0.00)	*0.63*	**0.13 (±0.00)**	** * **0.19** * **
	*Melaleuca alternifolia* (tea tree)	4.00 (±0.00)	1.17	**1.00 (±0.00)**	*0.58*	2.00 (±0.00)	*0.63*	1.50 (±0.71)	*0.75*	**1.00 (±0.00)**	*0.25*	3.00 (±1.41)	1.50	4.00 (±0.00)	1.50	2.00 (±0.00)	*0.63*	**0.09 (±0.00)**	** *0.16* **
	*Salvia sclarea* (clary sage)	2.00 (±0.00)	* **0.46** *	2.00 (±0.00)	*1.00*	2.00 (±0.00)	* **0.25** *	4.00 (±0.00)	1.33	2.00 (±0.00)	* **0.50** *	2.00 (±0.00)	*0.67*	**1.00 (±0.00)**	* **0.31** *	2.00 (±0.00)	*0.25*	**0.25 (±0.00)**	1.25
*Elettaria cardamonum* (cardamom)	*Coriandrum sativum* (coriander)	**0.50 (±0.00)**	*0.75*	**0.25 (±0.00)**	* **0.19** *	**0.50 (±0.00)**	* **0.38** *	**0.75 (±0.35)**	1.69	**1.00 (±0.00)**	* **0.31** *	**0.50 (±0.00)**	*0.75*	**1.00 (±0.00)**	1.50	2.00 (±0.00)	1.17	**0.13 (±0.00)**	2.04
*Eucalyptus globulus* (eucalyptus)	*Cymbopogon citratus* (lemongrass)	**1.00 (±0.00)**	*0.63*	**1.00 (±0.00)**	* **0.50** *	2.00 (±0.00)	1.67	**0.13 (±0.00)**	* **0.20** *	**1.00 (±0.00)**	* **0.38** *	3.00 (±1.41)	1.50	8.00 (±0.00)	2.50	2.00 (±0.00)	*1.00*	**0.06 (±0.00)**	** *0.04* **
*Ferula galbaniflua* (galbanum)	*Zingiber officinale* (ginger)	2.00 (±0.00)	1.33	2.00 (±0.00)	1.50	2.00 (±0.00)	*1.00*	**0.25 (±0.00)**	* **0.21** *	2.00 (±0.00)	*0.67*	8.00 (±0.00)	1.83	**1.00 (±0.00)**	* **0.38** *	2.00 (±0.00)	1.50	**0.25 (±0.00)**	*0.58*
*Helichrysum italicum* (immortelle)	*Lavandula angustifolia* (lavender)	2.00 (±0.00)	1.25	**1.00 (±0.00)**	* **0.50** *	2.00 (±0.00)	1.50	**0.75 (±0.35)**	* **0.34** *	2.00 (±0.00)	*1.00*	**1.00 (±0.00)**	* **0.29** *	**1.00 (±0.00)**	* **0.42** *	3.00 (±1.41)	1.75	**0.38 (±0.18)**	1.69
	*Lavandula spica* (lavender spike)	**1.00 (±0.00)**	*0.63*	**1.00 (±0.00)**	* **0.50** *	2.00 (±0.00)	1.17	**0.50 (±0.00)**	* **0.25** *	2.00 (±0.00)	*0.75*	6.00 (±2.83)	1.50	**0.75 (±0.35)**	* **0.31** *	2.00 (±0.00)	1.67	**0.50 (±0.00)**	2.17
*Hyssopus officinalis* (hyssop)	*Rosmarinus officinalis* (rosemary)	2.00 (±0.00)	*0.75*	2.00 (±0.00)	*1.00*	3.00 (±1.41)	1.50	**1.00 (±0.00)**	*1.00*	1.50 (±0.71)	*0.38*	4.00 (±0.00)	* **0.50** *	4.00 (±0.00)	2.00	2.00 (±0.00)	*1.00*	**0.09 (±0.00)**	*0.56*
*Laurus nobilis* (bay)	*Eucalyptus globulus* (eucalyptus)	**1.00 (±0.00)**	*0.63*	4.00 (±0.00)	2.00	8.00 (±0.00)	3.67	**0.63 (±0.04)**	* **0.26** *	2.00 (±0.00)	* **0.50** *	**1.00 (±0.00)**	* **0.29** *	**1.00 (±0.00)**	*0.58*	2.00 (±0.00)	*0.63*	**0.19 (±0.00)**	** *0.12* **
*Lavandula angustifolia* (lavender)	*Citrus aurantifolia* (lime)	3.00 (±1.41)	1.88	2.00 (±0.00)	*1.00*	2.00 (±0.00)	1.50	2.00 (±0.00)	*0.75*	**1.00 (±0.00)**	* **0.38** *	2.00 (±0.00)	* **0.46** *	**1.00 (±0.00)**	*0.58*	2.00 (±0.00)	*1.00*	**0.25 (±0.00)**	** *0.29* **
*Myrtus communis* (myrtle)	*Melaleuca alternifolia* (tea tree)	2.00 (±0.00)	* **0.50** *	**1.00 (±0.00)**	*0.58*	**1.00 (±0.00)**	* **0.50** *	2.00 (±0.00)	*0.79*	2.00 (±0.00)	* **0.42** *	4.00 (±0.00)	1.67	**1.00 (±0.00)**	* **0.38** *	**1.00 (±0.00)**	* **0.50** *	**0.25 (±0.00)**	** *0.23* **
*Ocimum basilicum* (basil)		**1.00 (±0.00)**	1.13	2.00 (±0.00)	1.67	2.00 (±0.00)	*1.00*	**0.63 (±0.00)**	* **0.42** *	**1.00 (±0.00)**	* **0.25** *	**0.75 (±0.35)**	* **0.31** *	1.50 (±0.71)	*0.94*	2.00 (±0.00)	1.50	**0.13 (±0.00)**	** *0.33* **
*Origanum marjorana* (marjoram)		2.00 (±0.00)	* **0.38** *	**1.00 (±0.00)**	*0.58*	1.50 (±0.71)	*0.75*	2.00 (±0.00)	1.67	3.00 (±1.41)	*0.88*	4.00 (±0.00)	2.00	**1.00 (±0.00)**	* **0.38** *	**1.00 (±0.00)**	* **0.50** *	**0.25 (±0.00)**	** *0.23* **
*Origanum vulgare* (origanum)		2.00 (±0.00)	1.25	**1.00 (±0.00)**	*0.58*	**1.00 (±0.00)**	* **0.38** *	2.00 (±0.00)	1.67	2.00 (±0.00)	* **0.50** *	4.00 (±0.00)	2.00	**1.00 (±0.00)**	* **0.38** *	3.00 (±1.41)	1.50	**0.50 (±0.00)**	*0.58*
*Salvia sclarea* *(clary* sage)	*Citrus aurantifolia* (lime)	4.00 (±0.00)	0.75	2.00 (±0.00)	1.00	2.00 (±0.00)	0.63	2.00 (±0.00)	0.83	1.50 (±0.71)	* **0.38** *	1.50 (±0.71)	* **0.34** *	8.00 (±0.00)	2.50	2.00 (±0.00)	0.63	**0.19 (±0.00)**	0.88
	*Citrus bergamia* (bergamot)	8.00 (±0.00)	1.00	**1.00 (±0.00)**	* **0.50** *	4.00 (±0.00)	1.25	8.00 (±0.00)	2.67	1.50 (±0.71)	* **0.38** *	1.50 (±0.71)	* **0.50** *	8.00 (±0.00)	1.83	2.00 (±0.00)	* **0.25** *	**0.13 (±0.00)**	0.56
	*Lavandula burnati* (lavandin)	**1.00 (±0.00)**	** *0.40* **	2.00 (±0.00)	1.00	2.00 (±0.00)	1.13	2.00 (±0.00)	1.33	2.00 (±0.00)	0.75	2.00 (±0.00)	0.83	2.00 (±0.00)	1.13	2.00 (±0.00)	0.63	**0.19 (±0.00)**	0.81
*Styrax benzoin* (benzoin)	*Mentha piperita* (peppermint)	**1.00 (±0.00)**	0.63	2.00 (±0.00)	1.00	2.00 (±0.00)	** *0.50* **	**1.00 (±0.00)**	0.67	**1.00 (±0.00)**	* **0.25** *	2.00 (±0.00)	1.00	**1.00 (±0.00)**	0.75	4.00 (±0.00)	2.00	**0.13 (±0.00)**	** *0.21* **
	*Rosa damascena* (rose)	2.00 (±0.00)	0.75	**1.00 (±0.00)**	** *0.50* **	**1.00 (±0.00)**	** *0.38* **	**0.50 (±0.00)**	2.17	1.50 (±0.71)	0.56	**0.50 (±0.00)**	* **0.38** *	2.00 (±0.00)	1.00	**1.00 (±0.00)**	* **0.50** *	**0.50 (±0.00)**	1.17
*Syzygium caryophyllata* (clove)	*Juniperus virginiana* (cederwood)	**0.50 (±0.00)**	0.63	**0.50 (±0.00)**	** *0.50* **	**1.00 (±0.00)**	0.75	**0.19 (±0.00)**	0.55	**1.00 (±0.00)**	** *0.50* **	**0.38 (±0.18)**	1.13	**1.00 (±0.00)**	* **0.50** *	2.00 (±0.00)	2.00	**0.13 (±0.00)**	1.13
	*Melaleuca alternifolia* (tea tree)	**1.00 (±0.00)**	** *0.38* **	**1.00 (±0.00)**	0.83	**1.00 (±0.00)**	0.75	**0.50 (±0.00)**	* **0.29** *	**1.00 (±0.00)**	** *0.38* **	**0.50 (±0.00)**	1.17	**1.00 (±0.00)**	* **0.38** *	2.00 (±0.00)	1.50	**0.06 (±0.00)**	0.54
	*Thymus vulgaris* (thyme)	**1.00 (±0.00)**	** *0.38* **	**0.50 (±0.00)**	** *0.50* **	**0.50 (±0.00)**	** *0.50* **	**1.00 (±0.00)**	1.25	4.00 (±0.00)	1.50	**0.50 (±0.00)**	1.50	**0.50 (±0.00)**	** *0.25* **	2.00 (±0.00)	2.00	**0.13 (±0.00)**	1.33
Positive control (ciprofloxacin)		0.50 × 10^−3^		0.50 × 10^−3^		0.50 × 10^−3^		0.50 × 10^−3^		0.50 × 10^−3^		0.25 × 10^−3^		1.00 × 10^−3^		0.50 × 10^−3^		n.a.	
Positive control (amphotericin b)		n.a.		n.a.		n.a.		n.a.		n.a.		n.a.		n.a.		n.a.		0.50 × 10^−3^	
Negative control (acetone in water)		>8.00		>8.00		>8.00		>8.00		>8.00		>8.00		>8.00		>8.00		>8.00	

MIC * denotes noteworthy antimicrobial effect (MIC less than or equal to 1 mg/mL); ƩFIC ** in bold and italics denotes synergistic antimicrobial effect (ƩFIC less than or equal to 0.50); while ƩFIC in italics only denotes additive antimicrobial effect (ƩFIC greater than 0.50 and less than or equal to 1.00).

**Table 2 antibiotics-10-01517-t002:** The mean percentage brine-shrimp viability after 48 h (*n* = 3) with standard deviation in brackets and ΣFIC values of the essential oil combinations investigated.

Essential Oil Combinations		Individual Brine-Shrimp Mortality (%) *		Combined Brine-Shrimp Mortality (%) *	ƩFIC **
Essential Oil 1	Essential Oil 2	Essential Oil 1	Essential Oil 2		
*Amyris balsamifera* (amyris)	*Styrax benzoin* (benzoin)	**1.4 (±0.89)**	**2.47 (±1.08)**	**5.65 (±4.78)**	3.16
*Citrus bergamia* (bergamot)	*Cupressus sempervirens* (cypress)	**1.22 (±1.55)**	**1.02 (±2.96)**	**0.65 (±1.13)**	*0.59*
*Cupressus sempervirens* (cypress)	*Hyssopus officinalis* (hyssop)	**1.02 (±2.96)**	**6.65 (±4.54)**	**8.80 (±4.98)**	4.99
	*Lavandula angustifolia* (lavender)		**9.59 (±3.23)**	**39.18 (±3.98)**	21.29
	*Melaleuca alternifolia* (tea tree)		**1.07 (±0.56)**	**8.87 (±1.71)**	8.48
	*Salvia sclarea* (clary sage)		**0.20 (±0.35)**	**11.81 (±5.88)**	34.67
*Elettaria cardamonum* (cardamom)	*Coriandrum sativum* (coriander)	**0.00 (±0.00)**	**11.95 (±2.96)**	**35.07 (±9.30)**	1754.74
*Ferula galbaniflua* (galbanum)	*Zingiber officinale* (ginger)	**15.77 (±0.09)**	**0.19 (±0.33)**	**28.08**	74.18
*Helichrysum italicum* (immortelle)	*Lavandula angustifolia* (lavender)	**9.50 (±4.54)**	**9.59 (±3.23)**	**13.45 (±11.74)**	1.41
	*Lavandula spica* (lavender spike)		**1.01 (±0.47)**	**26.38 (±5.63)**	14.46
*Hyssopus officinalis* (hyssop)	*Rosmarinus officinalis* (rosemary)	**6.65 (±4.54)**	**0.71 (±0.73)**	**13.07 (±1.68)**	10.23
*Laurus nobilis* (bay)	*Eucalyptus globulus* (eucalyptus)	**3.04 (±1.50)**	**4.21 (±1.09)**	**17.86 (±1.73)**	5.06
*Lavandula angustifolia* (lavender)	*Citrus aurantifolia* (lime)	**9.59 (±3.23)**	**0.36 (±0.62)**	**33.87 (±2.42)**	48.51
*Myrtus communis* (myrtle)	*Melaleuca alternifolia* (tea tree)	**0.00 (±0.00)**	**1.07 (±0.56)**	**28.33 (±4.76)**	1429.46
*Ocimum basilicum* (basil)		**13.04 (±6.25)**	**1.07 (±0.56)**	**39.69 (±2.34)**	19.99
*Origanum marjorana* (marjoram)		**0.34 (±6.25)**	**1.07 (±0.56)**	**6.03 (±1.85)**	11.76
*Origanum vulgare* (origanum)		99.67 (±0.57)	**1.07 (±0.56)**	**7.75 (±2.49)**	3.65
*Salvia sclarea* (clary sage)	*Citrus aurantifolia* (lime)	**0.20 (±0.35)**	**0.36 (±0.62)**	56.47 (±5.09)	80.06
	*Citrus bergamia* (bergamot)		**1.22 (±1.55)**	**15.76 (±5.97)**	128.15
	*Lavandula burnati* (lavandin)		**0.16 (±0.26)**	**12.64 (±3.46)**	279.78
*Styrax benzoin* (benzoin)	*Mentha piperita* (peppermint)	**2.47 (±1.08)**	**6.98 (±1.07)**	**3.31 (±1.09)**	12.28
	*Rosa damascena* (rose)		**97.32 (±2.34)**	**3.11 (±1.76)**	2.92
*Syzygium caryophyllata* (clove)	*Cinnamonum zeylanicum* (cinnamon)	99.57 (±0.74)	**98.49 (±0.52)**	**6.58 (±1.57)**	** *0.03* **
	*Cymbopogon citratus* (lemongrass)		**100 (±0.00)**	**2.30 (±1.24)**	** *0.05* **
	*Juniperus virginiana* (cederwood)		**0.57 (±0.23)**	**3.06 (±1.05)**	3.91
	*Thymus vulgaris* (thyme)		**97.76 (±2.56)**	**25.16 (±4.52)**	** *0.03* **
Positive control (Potassium dichromate)		99.95 (±0.03)			
Negative control (Sea water)		0.00 (±0.00)			
Solvent control (DMSO)		0.00 (±0.00)			

Brine-shrimp viability (%) * in bold denotes non-toxic effect (% mortality less than 50%); ƩFIC ** in bold denotes synergistic effect (ƩFIC less than or equal to 0.50) and ƩFIC in italics denotes additive effect (ƩFIC greater than 0.50 and less than or equal to 1.00).

**Table 3 antibiotics-10-01517-t003:** The mean percentage NO production of RAW 264.7 macrophages (*n* = 3) with standard deviation in brackets and ΣFIC values after treatment with essential oil combinations.

Essential Oil Combination		Individual Essential Oil Nitrite Production (μM)		Combined Essential Oils Nitrite Production (μM) *	ƩFIC *** % NO Production
Essential Oil 1	Essential Oil 2	Essential oil 1	Essential oil 2		
*Amyris balsamifera* (amyris)	*Styrax benzoin* (benzoin)	1.95 (±0.02)	2.45 (±0.07)	*2.22 (±0.03)*	1.02
*Ocimum basilicum* (basil)	*Melaleuca alternifolia* (tea tree)	2.91 (±0.19)	4.06 (±0.23)	3.83 (±0.10)	1.13
*Laurus nobilis* (bay)	*Eucalyptus globulus* (eucalyptus)	5.43 (±0.38)	4.57 (±0.19)	4.04 (±0.05)	*0.81*
*Styrax benzoin* (benzoin)	*Mentha piperita* (peppermint)	2.45 (±0.07)	6.38 (±0.44)	3.75 (±0.05)	1.06
	*Rosa damascene* (rose)	2.45 (±0.07)	4.71 (±0.40)	** *2.08 (±0.01)* **	*0.65*
*Citrus bergamia* (bergamot)	*Cupressus sempervirens* (cypress)	2.17 (±0.10)	2.02 (±0.10)	** *2.05 (±0.00)* **	*0.98*
*Elettaria cardamonum* (cardamom)	*Coriandrum sativum* (coriander)	2.12 (±0.06)	1.93 (±0.02)	** *2.01 (±0.03)* **	*0.99*
*Salvia sclarea* (clary sage)	*Citrus bergamia* (bergamot)	2.02 (±0.02)	2.17 (±0.10)	** *2.07 (±0.03)* **	*0.99*
	*Lavandula burnati* (lavandin)	2.02 (±0.02)	3.61 (±0.05)	4.30 (±0.05)	1.66
	*Citrus aurantifolia* (lime)	2.02 (±0.02)	2.00 (±0.02)	** *2.02 (±0.05)* **	1.00
*Syzygium caryophyllata* (clove)	*Juniperus virginiana* (cedarwood)	1.95 (±0.02)	3.38 (±0.02)	** *2.01 (±0.03)* **	*0.81*
	*Melaleuca alternifolia* (tea tree)	1.95 (±0.02)	4.06 (±0.23)	** *2.07 (±0.03)* **	*0.79*
*Cupressus sempervirens* (cypress)	*Salvia sclarea* (clary sage)	2.02 (±0.10)	2.02 (±0.02)	** *2.00 (±0.05)* **	*0.99*
	*Hyssopus officinalis* (hyssop)	2.02 (±0.10)	3.59 (±0.07)	** *1.98 (±0.02)* **	*0.77*
	*Lavandula angustifolia* (lavender)	2.02 (±0.10)	2.99 (±0.38)	** *2.00 (±0.03)* **	*0.83*
	*Melaleuca alternifolia* (tea tree)	2.02 (±0.10)	4.06 (±0.23)	3.49 (±0.02)	1.30
*Ferula galbaniflua* (galbanum)	*Zingiber officinale* (ginger)	1.99 (±0.03)	2.85 (±0.04)	** *2.34 (±0.03)* **	1.00
*Hyssopus officinalis* (hyssop)	*Rosmarinus officinalis* (rosemary)	3.59 (±0.07)	4.73 (±0.18)	3.54 (±0.03)	*0.87*
*Helichrysum italicum* (immortelle)	*Lavandula angustifolia* (lavender)	2.06 (±0.02)	2.99 (±0.04)	** *2.04 (±0.02)* **	*0.84*
	*Lavandula spica* (lavender spike)	2.06 (±0.02)	4.68 (±0.17)	3.44 (±0.18)	1.20
*Lavandula angustifolia* (lavender)	*Citrus aurantifolia* (lime)	2.99 (±0.04)	2.00 (±0.02)	** *2.97 (±0.05)* **	1.24
*Origanum marjorana* (marjoram)	*Melaleuca alternifolia* (tea tree)	4.81 (±0.11)	4.06 (±0.23)	3.05 (±0.14)	*0.69*
*Myrtus communis* (myrtle)		3.07 (±0.13)	4.06 (±0.23)	* **2.97 (±0.01)** *	*0.85*
*Origanum vulgare* (origanum)		4.88 (±0.47)	4.06 (±0.23)	3.87 (±0.18)	*0.87*
Medium only		1.77 (±4.52)			
Positive control (LPS + Aminoguanidine)		2.58 (±2.39)			
Negative control (LPS + medium)		4.90 (±9.21)			

Combined essential oil NO production (%) * bold and in italics denotes anti-inflammatory effect (μM less than 2.97); ƩFIC *** in bold denotes synergistic effect (ƩFIC less than or equal to 0.50); ƩFIC in italics denotes additive effect (ƩFIC greater than 0.50 but less than or equal to 1.00).

**Table 4 antibiotics-10-01517-t004:** The mean percentage cell viability of RAW 264.7 macrophages (*n* = 3) with standard deviation in brackets and ΣFIC values after treatment with essential oil combinations.

Essential Oil Combination		Cell Viability against Individual Essential Oils (%)		Cell Viability against Combined Essential Oils (%) **	ƩFIC *** % Cell Viability
Essential Oil 1	Essential Oil 2	Essential Oil 1	Essential Oil 2		
*Amyris balsamifera* (amyris)	*Styrax benzoin* (benzoin)	8.11 (±4.52)	47.98 (±0.08)	7.90 (±4.05)	*0.57*
*Ocimum basilicum* (basil)	*Melaleuca alternifolia* (tea tree)	50.63 (±2.39)	66.15 (±2.07)	** *66.80 (±3.90)* **	1.16
*Laurus nobilis* (bay)	*Eucalyptus globulus* (eucalyptus)	58.49 (±9.20)	67.01 (±0.23)	** *74.91 (±0.15)* **	1.2
*Styrax benzoin* (benzoin)	*Mentha piperita* (peppermint)	47.98 (±0.08)	61.83 (±3.25)	58.45 (±0.09)	1.08
	*Rosa damascene* (rose)		60.04 (±3.54)	36.90 (±1.75)	*0.69*
*Citrus bergamia* (bergamot)	*Cupressus sempervirens* (cypress)	33.40 (±4.98)	7.98 (±0.33)	8.68 (±2.65)	*0.67*
*Elettaria cardamonum* (cardamom)	*Coriandrum sativum* (coriander)	8.11 (±3.77)	9.12 (±5.27)	8.11 (±2.65)	*0.94*
*Salvia sclarea* (clary sage)	*Citrus bergamia* (bergamot)	14.99 (±0.15)	33.40 (±4.98)	8.07 (±0.88)	**0.39**
	*Lavandula burnati* (lavandin)		71.04 (±0.15)	55.64 (±0.69)	2.25
	*Citrus aurantifolia* (lime)		12.3 (±4.82)	8.15 (±0.15)	*0.60*
*Syzygium caryophyllata* (clove)	*Juniperus virginiana* (cederwood)	23.83 (±3.70)	57.68 (±3.99)	7.98 (±0.09)	**0.24**
	*Melaleuca alternifolia* (tea tree)		66.15 (±2.07)	51.65 (±0.23)	1.47
*Cupressus sempervirens* (cypress)	*Salvia sclarea* (clary sage)	7.98 (±0.33)	14.99 (±0.15)	9.94 (±0.23)	*0.95*
	*Hyssopus officinalis* (hyssop)		63.14 (±0.09)	12.18 (±0.13)	*0.86*
	*Lavandula angustifolia* (lavender)		71.53 (±3.44)	9.08 (±0.23)	*0.63*
	*Melaleuca alternifolia* (tea tree)		66.15 (±2.07)	** *83.83 (±8.08)* **	5.88
*Ferula galbaniflua* (galbanum)	*Zingiber officinale* (ginger)	8.07 (±5.13)	8.11 (±1.63)	8.07 (±1.74)	*1.00*
*Hyssopus officinalis* (hyssop)	*Rosmarinus officinalis* (rosemary)	63.14 (±0.09)	77.64 (±0.84)	** *79.59 (±3.12)* **	1.14
*Helichrysum italicum* (immortelle)	*Lavandula angustifolia* (lavender)	7.94 (±0.09)	71.53 (±3.44)	8.27 (±0.48)	*0.58*
	*Lavandula spica* (lavender spike)		68.92 (±0.08)	** *72.06 (±1.93)* **	5.06
*Lavandula angustifolia* (lavender)	*Citrus aurantifolia* (lime)	71.53 (±3.44)	12.3 (±4.82)	** *68.84 (±0.09)* **	3.28
*Origanum marjorana* (marjoram)	*Melaleuca alternifolia* (tea tree)	57.84 (±0.28)	66.15 (±2.07)	** *90.14 (±0.16)* **	1.46
*Myrtus communis* (myrtle)		59.43 (±4.00)		** *78.00 (±4.67)* **	1.25
*Origanum vulgare* (origanum)		68.84 (±3.69)		** *86.31 (±0.58)* **	1.28
LPS		109.31 (±0.04)			
Positive control (Adenosine Guanine)		119.51 (±0.23)			
Negative control (LPS + medium)		100.00 (±0.18)			

Combined essential oil cell viability (%) ** bold and italics denotes non-toxic effect (%cell viability greater than 50%); ƩFIC *** in bold denotes synergistic effect (ƩFIC less than or equal to 0.50); ƩFIC in italics denotes additive effect (ƩFIC greater than 0.50 but less than or equal to 1.00).

**Table 5 antibiotics-10-01517-t005:** The mean percentage cell viability of A549 lung cancer cell line (*n* = 3) with standard deviation in brackets and ΣFIC values after treatment with essential oil combinations.

Essential Oil Combination		A549 Cell Viability * Following Individual Essential Oils (%)		A549 Cell Viability * Following Combined Essential Oils (%)	ƩFIC % Cell Viability **
Essential Oil 1	Essential Oil 2	Essential Oil 1	Essential Oil 2		
*Amyris balsamifera* (amyris)	*Styrax benzoin* (benzoin)	**94.78 (±0.08)**	**91.25 (±11.08)**	**87.95 (±0.09)**	*0.95*
*Ocimum basilicum* (basil)	*Melaleuca alternifolia* (tea tree)	**87.16 (±4.98)**	**86.01 (±3.90)**	**79.62 (±1.75)**	*0.92*
*Laurus nobilis* (bay)	*Eucalyptus globulus* (eucalyptus)	**89.55 (±2.89)**	**99.63 (±4.69)**	**96.14 (±2.65)**	1.02
*Styrax benzoin* (benzoin)	*Mentha piperita* (peppermint)	**91.25 (±11.08)**	**94.95 (±3.54)**	**95.22 (±2.65)**	1.02
	*Rosa damascena* (rose)		**87.84 (±5.17)**	**88.38 (±0.88)**	*0.99*
*Citrus bergamia* (bergamot)	*Cupressus sempervirens* (cypress)	**96.41 (±8.23)**	**100.00 (±0.13)**	**93.88 (±0.69)**	*0.96*
*Elettaria cardamonum* (cardamom)	*Coriandrum sativum* (coriander)	**84.15 (±0.08)**	**97.36 (±0.23)**	**89.47 (±0.15)**	*0.99*
*Salvia sclarea* (clary sage)	*Citrus bergamia* (bergamot)	**99.72 (±0.33)**	**96.41 (±8.23)**	**94.11 (±0.09)**	*0.96*
	*Cupressus sempervirens* (cypress)		**100.00 (±0.13)**	**98.28 (±0.23)**	*0.98*
	*Lavandula burnati* (lavandin)		**98.88 (±0.74)**	**87.41 (±2.30)**	*0.88*
	*Citrus aurantifolia* (lime)		**100.00 (±4.00)**	**82.00 (±0.13)**	*0.82*
*Syzygium caryophyllata* (clove)	*Juniperus virginiana* (cedarwood)	**99.36 (±10.45)**	**95.86 (±3.70)**	**99.55 (±0.23)**	1.02
	*Melaleuca alternifolia* (tea tree)		**86.01 (±3.90)**	**81.75 (±8.08)**	*0.89*
*Cupressus sempervirens* (cypress)	*Salvia sclarea* (clary sage)	**100.00 (±0.13)**	**99.72 (±0.33)**	**96.66 (±1.74)**	*0.97*
	*Hyssopus officinalis* (hyssop)		**93.14 (±3.44)**	**98.00 (±3.12)**	1.02
	*Lavandula angustifolia* (lavender)		**99.21 (±6.20)**	**98.41 (±0.48)**	*0.99*
	*Melaleuca alternifolia* (tea tree)		**86.01 (±3.90)**	**87.23 (±1.93)**	*0.94*
*Ferula galbaniflua* (galbanum)	*Zingiber officinale* (ginger)	**100.00 (±0.00)**	**95.85 (±0.15)**	**99.42 (±0.09)**	1.02
*Hyssopus officinalis* (hyssop)	*Rosmarinus officinalis* (rosemary)	**93.14 (±3.44)**	**89.15 (±4.68)**	**96.23 (±0.16)**	1.06
*Helichrysum italicum* (immortelle)	*Lavandula angustifolia* (lavender)	**99.05 (±3.44)**	**99.21 (±6.20)**	**98.18 (±4.67)**	*0.99*
	*Lavandula spica* (lavender spike)		**99.31 (±4.82)**	**97.58 (±0.58)**	*0.98*
*Lavandula angustifolia* (lavender)	*Citrus aurantifolia* (lime)	**99.21 (±6.20)**	**100.00 (±4.00)**	**80.38 (±2.90)**	*0.81*
*Origanum marjorana* (marjoram)	*Melaleuca alternifolia* (tea tree)	**96.52 (±3.79)**	**86.01 (±3.90)**	**89.88 (±3.11)**	*0.99*
*Myrtus communis* (myrtle)		**98.24 (±3.69)**		**80.73 (±9.14)**	*0.88*
*Origanum vulgare* (origanum)		**95.03 (±3.43)**		**91.17 (±0.86)**	1.01
Positive control (Melphalan)		66.37 (±0.26)			
Negative control (Medium control)		92.30 (±0.27)			

A549 cell viability (%) * denotes non-toxic effects (% cell viability greater than 50%); ƩFIC ** in bold denotes synergistic effect (ƩFIC less than or equal to 0.50); ƩFIC in italics denotes additive effect (ƩFIC greater than 0.50 but less than or equal to 1.00).

## Data Availability

The data used to support the findings of this study are included in this study.

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
