# Peer review of "Essential Oil Blends: The Potential of Combined Use for Respiratory Tract Infections"

_antibiotics, 2021, doi:10.3390/antibiotics10121517_

Round 1

Reviewer 1 Report

Comments to Authors:

In this manuscript, the authors investigate the potential activity of essential oil combinations for antimicrobial, anti-toxic and anti-inflammatory activities. They have already tested the commercial essential oil and want to study their combination as used in aromatherapy for the treatment of respiratory tract infections.

General comments:

  1. English has to been revised.
  2. In particular phrases at lines: 50-51; 64-66; 89-93 have to been carefully revised.
  3. No data is provided of the error mean. These have to be added. Consequently, also the significance of the differences must to be added.
  4. No indication on the statistic used to analyse the results is reported. To consider confident the data results they have to be statistically analysed.

Specific comments:

  1. Lines 122-125. Why the Authors have studied the essential oil effects on “the most commonly neglected species” and not in the most prevalent observed in airway infection?
  2. Lines 155-157. Why Author have decided to define “toxic” essential oil combination that reach the 50% of shrimp death. I think that is a very high limit. For me toxic is compound that induce at least 20 percent of toxicity.
  3. Lines 163-164. The authors concludes that some essential oil combinations are less toxic than the single essential oil and that “the essential oils show the potential to quench toxicity of individual essential oils when used in combination”. Which is the concentration of the individual essential oil considered? When the combinations are tested the individual oils are diluted of two times, so the authors have to consider it. From the text seem that the Authors have confronted the obtained results with the results obtained in another publication in which this dilution is not considered.
  4. Line 176. Even if MTT assay is used to test the cytotoxicity of compounds this assay measures the metabolic activity of cells so a reduction of color can be obtained also for stressed cell but that can be resolved. It is incorrect define the MTT assay as more sensitive assay to test compound cytotoxicity.
  5. Line 205. Why authors have arbitrarily decided to considered active each compound that produce a nitrite level < 4 mM?
  6. Lines 253-254. “All studies were undertaken in triplicate on consecutive days” how many times each experiment is repeated?
  7. Lines 297-310. Is DMSO alone used as control?

Author Response

Reviewer #1:

General comments:

English has to been revised. In particular phrases at lines: 50-51; 64-66; 89-93 have to been carefully revised.

Line 50-51: “Essential oils, a natural product, has been extensively studied with relevance to respiratory conditions.”

The English has been checked throughout the manuscript. Page 2, lines 48-53 has been amended to read “Essential oils comprise of volatile, aromatic and complex chemical compounds such as alcohols, aldehydes, esters, ethers, ketones, phenols and terpenes. These essential oils are distilled from plant parts and commonly employed in aromatherapy. Within the scientific literature, essential oils have been extensively studied with relevance to respiratory conditions. A plethora of studies are available where essential oils have been studies for antimicrobial purposes as well as for inhalation in traditional practices and in maintaining basic health conditions [6-11].”

Line 64-66: “Layman’s literature, from aroma therapeutic practice note extensively the use of essential oils in combination for improved antimicrobial, anti-oxidative, an-ti-inflammatory, as well as antihistaminic effects [15].”

This has been changed in the manuscript on page 2, lines 63-65 to read “Aroma-therapeutic literature notes extensively the use of essential oils in combination for improved antimicrobial, anti-oxidative, anti-inflammatory, as well as antihistaminic effects [15]. Previous studies have demonstrated the therapeutic potential of some commercial and indigenous essential oils when tested in combination [16-20].”

Line 89-93: “It is a result of this lack of data to substantiate the use of essential oils in combination for antimicrobial and anti-inflammatory effects, as well as the increased risk of toxicity when combined, that further research is needed to confirm the safety and potential for use of these natural products via the respiratory tract.”

This has been changed in the manuscript on page 2, lines 98-99 to read “It is a result of this lack of data to substantiate the use of essential oils in combination for antimicrobial and anti-inflammatory effects, as well as the increased risk of toxicity associated with essential use when combined, that further research is needed to confirm the therapeutic potential for use of essential oils via the respiratory tract.”

No data is provided of the error mean. These have to be added. Consequently, also the significance of the differences must to be added.

The standard deviation is provided in Table 1 and S1, Table 2, Table 3, Table 4 as well as Table 5. The results calculated for standard deviation prove reliable, accurate results. No significant deviations or errors are noted.

No indication on the statistic used to analyse the results is reported. To consider confident the data results they have to be statistically analysed.

An addition to the methods has been made on page 11, line 358-362 to include data analysis. Statistical analysis has been added to Section 2.3. (page 3, line 204-208) to validate the cut off point for NO production in macrophages as a measure of anti-inflammatory activity.

Specific comments:

Lines 122-125. Why the Authors have studied the essential oil effects on “the most commonly neglected species” and not in the most prevalent observed in airway infection?

The pathogens reported within the body of the manuscript consider the pathogens most commonly neglected in research when concerning respiratory tract infections. These same pathogens, Streptococcus pyogenes, Streptococcus pneumoniae and Haemophilus influenzae are the also most commonly isolated bacterial pathogens in respiratory tract infections (S. Leigh-de Rapper, S. F. van Vuuren, C&B 2020, 17, e2000062).

Lines 155-157. Why Author have decided to define “toxic” essential oil combination that reach the 50% of shrimp death. I think that is a very high limit. For me toxic is compound that induce at least 20 percent of toxicity.

The interpretation of Brine Shrimp assay associated toxicity is provided in the reference (Bussmann, 2011). This method for allocation of toxicity to samples of brine shrimp death of greater than or equal to 50% is well-applied in literature. Other studies using this interpretation include:

  • Khumalo, G.P.; Sadgrove, N.J.; Van Vuuren, S.F.; Van Wyk, B.-E. South Africa’s Best BARK Medicines Prescribed at the Johannesburg Muthi Markets for Skin, Gut, and Lung Infections: MIC’s and Brine Shrimp Lethality. Antibiotics202110, 681. https://doi.org/10.3390/antibiotics10060681
  • Hilmi, Y., Abushama, M.F., Abdalgadir, H. et al.A study of antioxidant activity, enzymatic inhibition and in vitro toxicity of selected traditional Sudanese plants with anti-diabetic potential. BMC Complement Altern Med 14, 149 (2014). https://doi.org/10.1186/1472-6882-14-149
  • Hlambani, S., Carmen, L., Geoffrey, C. et al. Antimicrobial activity and toxicity profile of selected southern African medicinal plants against neglected gut pathogens.  Afr. J. Sci.115 (11-12), 1-10. (2019). https://dx.doi.org/10.17159/sajs.2019/6199
  • Hübsch, Z., Van Zyl, R.L., Cock, I.E. et al. Interactive antimicrobial and toxicity profiles of conventional antimicrobials with Southern African medicinal plants. Afr. J. Bot., 93, 185-197 (2014).

Lines 163-164. The authors concludes that some essential oil combinations are less toxic than the single essential oil and that “the essential oils show the potential to quench toxicity of individual essential oils when used in combination”. Which is the concentration of the individual essential oil considered? When the combinations are tested the individual oils are diluted of two times, so the authors have to consider it. From the text seem that the Authors have confronted the obtained results with the results obtained in another publication in which this dilution is not considered.

The BSLA was used to quantify the toxic effects of the selected essential oils. A volume of 400 μL salt water containing on average 40–60 live brine-shrimp was added to each well of a 48 well micro-titre plate. Thereafter, 400 μL of essential oil sample (essential oil or a combination of essential oils (1 : 1), all diluted in 1% dimethyl sulphoxide (DMSO)) was added to wells. The combined essential oils were provided at 200 μL each when added to a single well. The toxicity was then measured based on this dilution.

The concern that the essential oils are double-diluted has been factored into the calculation and results and interpretation is given based on the results presented.

Line 176. Even if MTT assay is used to test the cytotoxicity of compounds this assay measures the metabolic activity of cells so a reduction of color can be obtained also for stressed cell but that can be resolved. It is incorrect define the MTT assay as more sensitive assay to test compound cytotoxicity.

Sentence reworded (line 176-179) to consider the metabolic measurement of the MTT assay.

Line 205. Why authors have arbitrarily decided to considered active each compound that produce a nitrite level < 4 mM?

Statisitcal analysis was undertaken using StatSoft Inc. (2004) STATISTICA (Data Analysis Software System), Version 7 software. An area under the curve (AUC) and Receiver Operator Characteristic (ROC) study was performed to determine the optimum cut-off point for the production of NO. This was created taking into consideration cell cytotoxicity. The active combinations were therefore determined to be 2.97 μM and the interpretation of this updated to reflect this change in Section 2.3, page 3, lines 204-208.

Lines 253-254. “All studies were undertaken in triplicate on consecutive days” how many times each experiment is repeated?

These experiments were repeated three times. This has been added to the methods of the manuscript (page 9, line 275-276).

Lines 297-310. Is DMSO alone used as control?

Yes. Added to the manuscript (page 11, lines 325-326) to reflect this is, “The negative control consisted of 32 g/L salt water, a solvent control of 1mg/mL DMSO in water and the positive control consisted of 1.60 mg/mL potassium dichromate (Fluka).”

Reviewer 2 Report

The work by Rapper and colleagues presents original findings of pharmacological relevance. Nevertheless, major revisions are required before the manuscript can be considered suitable for publication

While the manuscript is overall well-written, the results are poorly connected. So, the purposes, findings, discussion, and conclusion of the article should be restructured.

Introduction

Line 50: “Essential oils, a natural product,”… Here "a natural product" gives a very limited definition given the complexity and diversity of essential oils. Please review.

Line 66: Please, add “Therapeutic” before “Potential”

Lines 66-81. You should fragment this paragraph into clearly delimited sections describing: A) inflammation and its mechanisms/ physiological roles; B) Inflammation in the context of infection; C) The potential therapeutic roles of essential oils in modulating these processes.

Lines 72 and 73: “Inflammation, which is a complex process, leads to the release of cytokines and pro-inflammatory mediators that regulate the hosts’ response to infections”. Note that A) some cytokines are also pro-inflammatory mediators B) Inflammation may also occur in the absence of infection. Consider reviewing your statements.

While NO production was used as the only parameter for anti-inflammatory evaluation, its production, in particular by activated macrophages has not been mentioned in the introduction.

Results:

2.1 Reconsider your classification of the activity of the essential oils, as in your study, a MIC greater than 1 mg/ml is considered noteworthy while a MIC less than 1 is considered non-noteworthy. Rather, it would be relevant to classify the activity of the different oils as strong, weak, and moderate. For example, there are significant differences in the activity of essential oils that have MIC values equal to 0.25, 0.5, and 1 mg/ml.

2.2. Again, I think that the representation of toxicity in figure 2 is too simplified. What if a combination induces exactly 50% mortality? Consider finding another way to represent toxicity variation. In the discussion, it is strongly recommended commenting how toxicity is observed in oleos of the same genus/family. Maybe this is an interesting way of representing your results.

Figure 3 indicates that essential oil combinations inducing nitrite production at concentrations > 4 µM are inflammatory, while those with nitrite production at concentrations < 4 µM are anti-inflammatory.

You should seriously rethink it. First because while NO production can be used as a parameter of macrophage activation, it cannot be used to classify a so complex phenomenon as inflammation. Then, where is statistical significance considered when you use such classification?

Table 3. Please, indicate how statistical analysis has been performed.

What is the relevance of the results 2.4 for your work? How do these findings connect with the others? Please, discuss it.

Methods

A brief description of figure 4 is required to a better understanding of the study design

Line 316 -31. Be careful when affirming that LPS – iNOS - NO “cause inflammation 317 in RAW 264.7 macrophage cells”. Instead, such signaling cascade in macrophages may contribute to the inflammatory response.

Cytotoxicity against A549 cell lines is not described.

Author Response

Reviewer #2:

Introduction: Line 50: “Essential oils, a natural product,”… Here "a natural product" gives a very limited definition given the complexity and diversity of essential oils. Please review.

This has been changed in the manuscript on page 2, lines 49-53 to read “Essential oils comprise of volatile, aromatic and complex chemical compounds such as alcohols, aldehydes, esters, ethers, ketones, phenols and terpenes. These essential oils are distilled from plant parts and commonly employed in aromatherapy. Within the scientific literature, essential oils have been extensively studied with relevance to respiratory conditions. A plethora of studies are available where essential oils have been studies for antimicrobial purposes as well as for inhalation in traditional practices and in maintaining basic health conditions [6-11].”

Introduction: Line 66: Please, add “Therapeutic” before “Potential”

This has been changed in the manuscript on page 2, lines 64-65 to read “Previous studies have demonstrated the therapeutic potential of some commercial and indigenous essential oils when combined with each other and with carrier oils [16-20].”

Introduction: Lines 66-81. You should fragment this paragraph into clearly delimited sections describing: A) inflammation and its mechanisms/ physiological roles; B) Inflammation in the context of infection; C) The potential therapeutic roles of essential oils in modulating these processes.

An expansion on these topics has been undertaken and structured as suggested by the reviewer in the body of the manuscript. These additions are provided on page 2, lines 67-90.

Introduction: Lines 72 and 73: “Inflammation, which is a complex process, leads to the release of cytokines and pro-inflammatory mediators that regulate the hosts’ response to infections”. Note that A) some cytokines are also pro-inflammatory mediators B) Inflammation may also occur in the absence of infection. Consider reviewing your statements.

This statement has been amended to reflect the reviewers’ comments in lines 74-107.

While NO production was used as the only parameter for anti-inflammatory evaluation, its production, in particular by activated macrophages has not been mentioned in the introduction.

The role of activated macrophages in the production of NO has been provided for in the introduction (page 2, lines 73-75).

Results: 2.1 Reconsider your classification of the activity of the essential oils, as in your study, a MIC greater than 1 mg/ml is considered noteworthy while a MIC less than 1 is considered non-noteworthy. Rather, it would be relevant to classify the activity of the different oils as strong, weak, and moderate. For example, there are significant differences in the activity of essential oils that have MIC values equal to 0.25, 0.5, and 1 mg/ml.

The interpretation of noteworthy antimicrobial effects determined using MIC microdilution methods is a well-established interpretation, used widely in the microbiology community. Some other studies using this interpretation include:

  • Madikizela, B., Ndhlala, AR., Finnie, JF., Van Staden, J (2013) In vitro antimicrobial activity of extracts from plants used traditionally in South Africa to treat tuberculosis and related symptoms. Evidence-Based Complementary and Alternative Medicine, 2013, 840719, https://doi.org/10.1155/2013/840719
  • Makunga, NP., Colling, J., Horsthemke, HR., Ramogola, WPN, Van Staden J (2007) Changing the chemical mosaic of South African medicinal plants through biotechnological strategies. South African Journal of Botany, 73, 299
  • Asong JA, Amoo SO, McGaw LJ, Nkadimeng SM, Aremu AO, Otang-Mbeng W (2019) Antimicrobial activity, antioxidant potential, cytotoxicity and phytochemical profiling of four plants locally used against skin diseases. Plants (Basel), 15, 8(9):350. doi: 10.3390/plants8090350.

However, a sentence has been added to the manuscript in lines 130-132 to indicate how many of the combinations demonstrated strongly noteworthy antimicrobial activity (less than 0.50 mg/mL) and those that demonstrated moderately noteworthy antimicrobial effects (between 0.50 and 1.00 mg/mL).

Results: 2.2. Again, I think that the representation of toxicity in figure 2 is too simplified. What if a combination induces exactly 50% mortality? Consider finding another way to represent toxicity variation. In the discussion, it is strongly recommended commenting how toxicity is observed in oleos of the same genus/family. Maybe this is an interesting way of representing your results.

The interpretation of Brine Shrimp assay associated toxicity is provided in the reference (Bussmann, 2011). This method for allocation of toxicity to samples of brine shrimp death of greater than or equal to 50% is well-applied in literature. Other studies using this interpretation include:

  • Khumalo, G.P.; Sadgrove, N.J.; Van Vuuren, S.F.; Van Wyk, B.-E. South Africa’s Best BARK Medicines Prescribed at the Johannesburg Muthi Markets for Skin, Gut, and Lung Infections: MIC’s and Brine Shrimp Lethality. Antibiotics202110, 681. https://doi.org/10.3390/antibiotics10060681
  • Hilmi, Y., Abushama, M.F., Abdalgadir, H. et al.A study of antioxidant activity, enzymatic inhibition and in vitro toxicity of selected traditional Sudanese plants with anti-diabetic potential. BMC Complement Altern Med 14, 149 (2014). https://doi.org/10.1186/1472-6882-14-149
  • Hlambani, S., Carmen, L., Geoffrey, C. et al. Antimicrobial activity and toxicity profile of selected southern African medicinal plants against neglected gut pathogens.  Afr. J. Sci.115 (11-12), 1-10. (2019). https://dx.doi.org/10.17159/sajs.2019/6199
  • Hübsch, Z., Van Zyl, R.L., Cock, I.E. et al. Interactive antimicrobial and toxicity profiles of conventional antimicrobials with Southern African medicinal plants. Afr. J. Bot., 93, 185-197 (2014).

However, the further classification of toxicity was considered for addition to the manuscript to indicate non-toxic (0 to 48%) to borderline toxic (49 to 51%) and toxic (> 51%). This exercise proved no borderline toxicity when used individually with only three combinations demonstrating this effect with values of 49.36%, 49.47% and 49.86% thus considered non-toxic. This measure of toxicity is only one assay for assessment and as such, the manuscript then further considers the result of the MTT cytotoxicity on A549 and macrophage cells to provide further depth to the study.

In regards to how toxicity is observed in oleos of the same genus/family, one needs to consider that the same plant can display differing degrees of toxicity dependent on method of essential oil extraction, region of harvest, and even season of harvest as these factors influence essential oil chemistry. When compared to genus/family, these same variations exist and as such, oils from varying regions should not be compared based on genus.

Results: Figure 3 indicates that essential oil combinations inducing nitrite production at concentrations > 4 µM are inflammatory, while those with nitrite production at concentrations < 4 µM are anti-inflammatory. You should seriously rethink it. First because while NO production can be used as a parameter of macrophage activation, it cannot be used to classify a so complex phenomenon as inflammation. Then, where is statistical significance considered when you use such classification?

Statisitcal analysis was undertaken using StatSoft Inc. (2004) STATISTICA (Data Analysis Software System), Version 7 software. An area under the curve (AUC) and Receiver Operator Characteristic (ROC) study was performed to determine the optimum cut-off point for the production of NO. This was created taking into consideration cell cytotoxicity. The active combinations were therefore determined to be 2.97 μM and the interpretation of this updated to reflect this change in Section 2.3 (page 3, lines 204-308). In addition, the investigation of NO production may not solely classify inflammation, however, this assay is a good initial indicator of anti-inflammatory potential. Downstream targets such as iNOS, COX-2 and NFkappaB can be used to further investigate and confirm these results.

Table 3. Please, indicate how statistical analysis has been performed.

Statisitcal analysis was undertaken using StatSoft Inc. (2004) STATISTICA (Data Analysis Software System), Version 7 software. An area under the curve (AUC) and Receiver Operator Characteristic (ROC) study was performed to determine the optimum cut-off point for the production of NO. This was created taking into consideration cell cytotoxicity. The active combinations were therefore determined to be 2.97 μM and the interpretation of this updated to reflect this change in Section 2.3.

What is the relevance of the results 2.4 for your work? How do these findings connect with the others? Please, discuss it.

The study focuses on the use of essential oils in the respiratory tract, therefore an additional assessment of the cytotoxic effects of these oils was tested against lung tissue. These findings supported those determined in the BSLA and provide an initial indication for the safe therapeutic potential of essential oils when used in the respiratory tract. These results therefore helped to determine the combination of essential oils determined as antimicrobial in effect while non-toxic against brine shrimp, macrophages and A549 lung cells. This has been added to the manuscript (page 9, lines 264-265).

Methods: A brief description of figure 4 is required to a better understanding of the study design

A description relating to figure 4 has been added to the manuscript (page 9, line 272-276).

Methods: Line 316 -31. Be careful when affirming that LPS – iNOS - NO “cause inflammation 317 in RAW 264.7 macrophage cells”. Instead, such signaling cascade in macrophages may contribute to the inflammatory response.

Wording has been changed in the manuscript to address the use of NO production as an indicator of anti-inflammatory potential and not a direct measure of this (page 11, lines 337-338).

Methods: Cytotoxicity against A549 cell lines is not described.

Method is provided in 3.4. Anti-inflammatory and MTT Cytotoxicity Assay in lines 333-357.

Round 2

Reviewer 1 Report

I found the that the revisions are in accordance with the request.  

Reviewer 2 Report

The authors have addressed all revision requests and therefore, the paper is now suitable for publication.